# Data-Driven Microstructure Property Relations

**Julian Lißner**  **and Felix Fritzen ***

Efficient Methods for Mechanical Analysis, Institute of Applied Mechanics (CE), University of Stuttgart, 70569 Stuttgart, Germany; lissner@mechbau.uni-stuttgart.de
* Correspondence: fritzen@mechbau.uni-stuttgart.de; Tel.: +49-711-685 66283

**Abstract:** An image based prediction of the effective heat conductivity for highly heterogeneous microstructured materials is presented. The synthetic materials under consideration show different inclusion morphology, orientation, volume fraction and topology. The prediction of the effective property is made exclusively based on image data with the main emphasis being put on the 2-point spatial correlation function. This task is implemented using both unsupervised and supervised machine learning methods. First, a snapshot proper orthogonal decomposition (POD) is used to analyze big sets of random microstructures and, thereafter, to compress significant characteristics of the microstructure into a low-dimensional feature vector. In order to manage the related amount of data and computations, three different incremental snapshot POD methods are proposed. In the second step, the obtained feature vector is used to predict the effective material property by using feed forward neural networks. Numerical examples regarding the incremental basis identification and the prediction accuracy of the approach are presented. A Python code illustrating the application of the surrogate is freely available.

**Keywords:** microstructure property linkage; unsupervised machine learning; supervised machine learning; neural network; snapshot proper orthogonal decomposition

**MSC:** 74-04, 74A40, 74E30, 74Q05, 74S30

## 1. Introduction

In material analysis and design of heterogeneous materials, multiscale modeling can be used for the discovery of microstructured materials with tuned properties for engineering applications. Thereby, it contributes to the improvement of the technical capabilities, reduces the amount of resources invested into the construction and enhances the reliability of the description of the material behavior. However, the discovery of materials with the desired material property, which is characterized by the microstructure of the solid, constitutes a highly challenging inverse problem.

The basis for all multiscale models and simulations is information on the microstructure and on the microscale material behavior. If at hand, physical experiments can be replaced by—often costly—computations in order to determine the material properties by virtual testing [1–3]. Separation of structural and microstructural length scales can often be assumed. This enables the use of the representative volume element (RVE) [4] equipped with the preferable periodic fluctuation boundary conditions [5]. The RVE characterizes the highly heterogeneous material using a single frame (or image) and the (analytical or numerical) computation can be conducted on this frame.

The concurrent simulation of the underlying microstructure (e.g., through nested FE simulations, cf., e.g., [6,7], or considering microstructure behavior in the constitutive laws, e.g., [8]) and of the problem on the structural scale is computationally intractable. In view of the correlation between computational complexity and energy consumption, nested FE simulations should be limited in application for ecological reasons, too. Therefore, efficient methods giving reliable prediction of

the material property are an active field of research: POD-driven reduced order models with hyper-reduction (e.g., [9,10]), with multiple reduced bases spanning also internal variable [11,12] and for finite strains [13,14] are a selection of recent examples. We refer also to general review articles on the topic such as [15,16].

Supposing that two similar images representing microstructured materials are considered, it is natural to expect similar effective properties in many physically relevant problems such as elasticity, thermal and electric conduction to mention only two applications. The main task, thus, persists in finding low-dimensional parameterizations of the images that capture the relevant information, use these parameterizations to compress the image information and build a surrogate model operating only on the reduced representation. A black-box approach, exploiting precomputed data for the construction of the surrogate to link features to characteristics and using established machine learning methods, is the topic of this paper.

As the no free lunch theorem [17] states, an algorithm can not be arbitrarily fast and arbitrarily accurate at the same time. Hence, there has to be a compromise either in accuracy, computational speed or in versatility. At the cost of generality, i.e., by focusing on subclasses of microstructures, fast and accurate models can be deployed while still allowing for considerable variations of the microstructures. This does not mean that these subclasses must be overly confined: For instance, inclusion volume fractions ranging from 20 up to 80% are considered in this work. Using a limited number of computations performed on relevant microstructure images, machine learned methods can be trained for the subclass under consideration. The sampling of the data, the feature extraction and the training of the machine learning (ML) algorithm constitutes the offline phase in which the surrogate model is built. Typically, the evaluation of the surrogate can be realized almost in real-time (at least this is the aspired and ambitious objective), thereby enabling previously infeasible applications in microstructure tayloring, interactive user interfaces and computations on mobile devices.

To have a reliable prediction for a broader range of considered microstructures, the material knowledge system (MKS) framework [18] is currently actively researched. Many branches thereof exist, all trying to attain low-dimensional microstructure descriptors from the truncation of selected $n$-point correlation functions. For instance, a principal component analysis (PCA) of the 2-point correlation functions is performed, using the principal scores in a polynomial regression model, in order to predict material properties. The MKS is actively researched for different material structures [19–21]. For instance, [19,20] successfully predict the elastic strain and yield stress for the underlying microstructure using the MKS approach, however they confine their focus on either the topological features of the microstructure or a confined range of allowed volume fractions (0–20%), often held constant in individual studies.

A different approach for target driven microstructure tayloring deploys reconstruction techniques [22,23] to generate similar microstructures which fulfill certain criteria. In order to explicitly find the optimal microstructure geometry, sensitivities of descriptors, as, e.g., the number of inclusions, with respect to material property are obtained with machine learning [24,25]. With the sensitivities at hand, target driven construction enables the generation of optimal microstructure topology for the desired material property, even when considering a broad design space [26].

The goal of the present study is to make accurate image based predictions for RVEs spanning large subclasses of all possible microstructured materials: Substantial variations of the volume fraction, the morphology and of the topology are considered.

Similarly to key ideas of the MKS approach, a reduced basis is deployed to reduce the dimensionality of the microstructural features contained in the $n$-point correlation functions. With the sheer amount of samples required, conventional methods fail to capture the key features of all considered microstructures. Therefore, we propose three novel incremental reduced basis updates to make the computation possible. Combining these techniques with the use of synthetic microstructure data, the costly training of the reduced basis and of the artificial neural network (e.g., [27]) become feasible, thereby allowing the creation of a surrogate model for the image-property linkage. The

surrogate accepts binarized image representations of bi-phasic materials as inputs. The outputs constitute the effective heat conductivity tensor of the considered material.

In Section 2 the microstructure classification and the three different incremental snapshot POD procedures used during feature extraction are presented (unsupervised learning). In Section 3 the use of feedforward artificial neural networks for the processing of the extracted features is discussed. Numerical examples are presented in Section 4 including different inclusion morphologies and an investigation of the relaxation of the microstructure subclass confinement, of the procedure by using mixed data sets, is made. A Python code illustrating the application of the surrogate is freely available via Github.

## 2. Materials and Methods

### 2.1. Microstructure Classification

The microstructure is defined by the representative volume element (RVE) [4], which is one periodic frame (or image) characterizing the heterogeneous material under consideration, see Figure 1 for examples of the microstructure and its 2-point spatial correlation function (see below for its definition). Due to their favorable properties regarding the needed size of the RVE, periodic fluctuation boundary conditions, e.g., [5], are used for the computations during the offline phase.

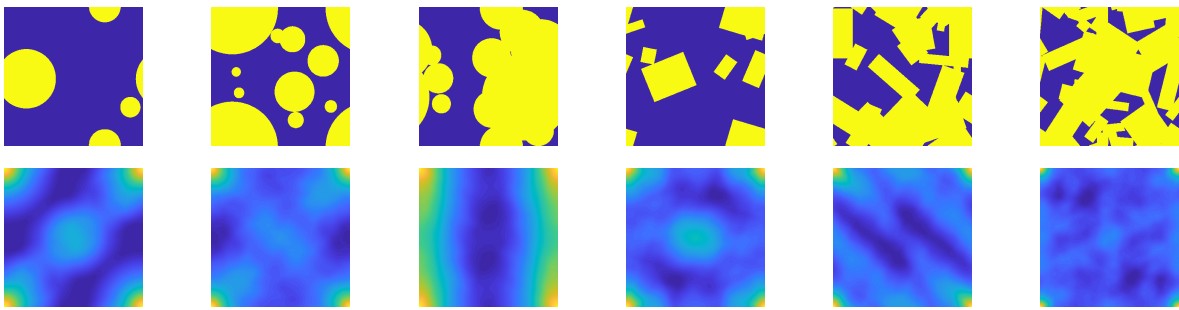

**Figure 1.** Depicting some exemplary microstructures with their respective 2-point spatial correlation functions $c_2(\mathbf{r}; b, b)$ below.

The n-point spatial correlation functions represent a widely used mathematical framework for microstructural characterization [28,29]. Roughly described, the n-point correlation is obtained by placing a polyline consisting of $(n-1)$ nodes defined relative to the first point by vectors $\mathbf{r}_1, \mathbf{r}_2, \ldots$.

By placing the first point uniformly randomly into the microstructure and computing the mean probability of finding a prescribed sequence of material phases at the nodes of the polyline (including the initial point) denotes the n-point correlation $c_n(\mathbf{r}_1, \mathbf{r}_2, \ldots, \mathbf{r}_{n-1}; m_1, m_2, \ldots, m_n)$, where $m_k$ is the material label expected to be found at the $k$th node.

For example, the 1-point spatial correlation function, i.e., the probability of finding phase $m$ ($m \in \{a, b, \ldots\}$), yields the phase volume fraction $f_m$ of phase $m$. In the present study bi-phasic materials are considered. Here $m = a$ corresponds to the matrix material (drawn blue in Figure 1) and $m = b$ to the inclusion phase (drawn yellow in Figure 1). The trivial relation

$$f_a = 1 - f_b \tag{1}$$

holds. The 2-point spatial correlation function (2PCF) $c_2(\mathbf{r}; a, b)$ places the vector $\mathbf{r}$ in each pixel/voxel $\mathbf{x}$ of the RVE and states the probability of starting in the matrix phase $a$ and ending in the inclusion phase $b$. Mathematically we have

$$c_2(\mathbf{r}; a, b) = \left\langle \chi^{(a)}(\mathbf{x}) \, \chi^{(b)}(\mathbf{x} + \mathbf{r}) \right\rangle_x \tag{2}$$

with $\chi^{(m)}$ being the indicator function of phase $m$, $\boldsymbol{r}$ the point offset and $\langle \bullet \rangle_x$ denoting the averaging operator over the RVE. The 2PCF is efficiently computed in Fourier space by making use of the algorithmically sleek fast Fourier transform (FFT) [30,31]

$$c_2(\boldsymbol{r}; a, b) = \mathscr{F}^{-1}\left(\overline{\mathscr{F}(\chi^{(a)})} \odot \mathscr{F}(\chi^{(b)})\right), \tag{3}$$

where $\mathscr{F}$ and $\mathscr{F}^{-1}$ denote the forward and backward FFT, $\overline{\bullet}$ is the complex conjugate and $\odot$ denotes the point-wise multiplication, respectively. For bi-phasic materials the three different two-point functions $c_2(\bullet; a, b), c_2(\bullet; a, a), c_2(\bullet; b, b)$ are related via

$$c_2(\boldsymbol{r}; a, a) = f_a - c_2(\boldsymbol{r}; a, b), \qquad\qquad c_2(\boldsymbol{r}; b, b) = f_b - c_2(\boldsymbol{r}; a, b). \tag{4}$$

In view of computational efficiency, this redundancy can be exploited. Some key characteristics of the non-negative 2PCF are

$$c_2(\boldsymbol{0}; a, a) = f_a = \max_{\boldsymbol{r} \in \Omega} c_2(\boldsymbol{r}; a, a), \tag{5}$$

$$c_2(\boldsymbol{0}; b, b) = f_b = \max_{\boldsymbol{r} \in \Omega} c_2(\boldsymbol{r}; b, b), \tag{6}$$

$$c_2(\boldsymbol{0}; a, b) = 0, \tag{7}$$

$$c_2(\boldsymbol{r}; a, b) = c_2(\boldsymbol{r}; b, a) = c_2(-\boldsymbol{r}; a, b), \tag{8}$$

$$\langle c_2(\boldsymbol{x}; m, m)\rangle_x = f_m^2 \qquad\qquad (m = a, b). \tag{9}$$

In addition to that, a key property of the 2PCF is its invariance with respect to translations of the periodic microstructure. This property is of essential importance when it comes to the comparison of several images under consideration, i.e., during the evaluation of similarities within images.

Examples of $c_2(\boldsymbol{r}; b, b)$ (referred to also as auto-correlation of the inclusion phase) are depicted by the lower set of images in Figure 1. By the metric of vision, the following characteristics can be observed:

- The maximum of $c_2(\boldsymbol{r}; b, b)$ occurs at the corners of the domain (corresponding to $\boldsymbol{r} = \boldsymbol{0}$);
- Preferred directions of the inclusion placement and/or orientation correspond to laminate-like images (best seen in the third microstructure from the left);
- The domain around $\boldsymbol{r} = \boldsymbol{0}$ partially reflects the average inclusion shape;
- Some similarities are found, particularly with respect to shape of the 2PCF at the corners and in the center.

These observations hint at the existence of a low-dimensional parameterization of relevant microstructural features. In the following this property is exploited by using a snapshot proper orthogonal decomposition (snapshot POD) in order to capture reoccurent patterns of the 2PCF. By working on the two-point function the afore-mentioned elimination of possible translations of the images is an important feature.

The influence of higher order spatial correlation functions has been investigated in the literature, e.g., [28,32]. These considerations often yield minor gains relative to the additional computations and the increased dimensionality (for instance, the 3PCF takes to vectors $\boldsymbol{r}_1, \boldsymbol{r}_2 \in \Omega$ as inputs. Hence, the full 3PCF is basically inaccessible in practice but only after major truncation). While it has been demonstrated that the two point function does not suffice to uniquely describe the microstructure in periodic domains [33], there is evidence that the level of microstructural ambiguity for identical 2PCF can be considered low. Therefore, only the n-point correlation functions up to second order are accounted for in the present study.

### 2.2. Unsupervised Learning via Snapshot Proper Orthogonal Decomposition

The snapshot POD [34] can be used to construct a reduced basis (RB) [35–37] that provides an optimal subspace for approximating a given snapshot matrix $\underline{\underline{S}} \in \mathbb{R}^{n \times n_s}$. The matrix $\underline{\underline{S}}$ consists of $n_s$ individual snapshots $\underline{s}_i \in \mathbb{R}^n$ with the size $n$ being the dimension of the discrete representation of the unreduced field information. In the case of the 2PCF $n$ denotes the total number of pixels within the RVE, i.e., the discrete two-dimensional 2PCF (represented as image data) is recast into vector format for further processing ($\underline{c}_2^0(m, m) \in \mathbb{R}^n$). In the present study, the constructed RB is used for information compression, i.e., for the extraction of relevant microstructural features from the image data. The reduced basis $\underline{\underline{B}} \in \mathbb{R}^{n \times N}$ retains the $N$ most salient features of the data contained in $\underline{\underline{S}}$ in a few eigenmodes represented by the orthonormal columns of $\underline{\underline{B}}$.

The actual snapshot data stored in $\underline{\underline{S}}$ is constructed from the discrete 2-point function data $\underline{s}_i^0 \in \mathbb{R}^n$ via scaling and shifting according to

$$\underline{s}_i = \frac{1}{f_b} \left( \underline{s}_i^0 - f_b^2 \underline{1} \right),$$

(10)

where $\underline{1} \in \mathbb{R}^n$ is a vector containing ones at all entries. This shift ensures a peak value of 1 in the corner and the mean of 0 for every snapshot.

The reduced basis is computed under the premise to minimize the overall relative projection error

$$\mathcal{P}_\delta = \frac{||\underline{\underline{S}} - \underline{\underline{B}}\,\underline{\underline{B}}^\mathsf{T}\,\underline{\underline{S}}||_F}{||\underline{\underline{S}}||_F}$$

(11)

with respect to the Frobenius norm $|| \bullet ||_F$. The RB can be constructed with multiple methods, e.g., with the snapshot correlation matrix $\underline{\underline{C}}_S$ and its eigenvalue decomposition, which is given by

$$\underline{\underline{C}}_S = \underline{\underline{S}}^\mathsf{T}\,\underline{\underline{S}} = \underline{\underline{V}}\,\underline{\underline{\Theta}}\,\underline{\underline{V}}^\mathsf{T}.$$

(12)

The following properties of the sorted eigenvalue decomposition hold

$$\underline{\underline{V}}^\mathsf{T}\,\underline{\underline{V}} = \underline{\underline{I}} \qquad \mathbb{R}^{n_s \times n_s}, \qquad\qquad \Theta_{ij} = \theta_i \delta_{ij}, \qquad\qquad \theta_1 \geq \theta_2 \geq ... \geq \theta_{n_s} \geq 0,$$

(13)

and $\delta_{ij}$ denotes the Kronecker delta. The dimension of the reduced basis is determined by the POD threshold, i.e., the truncation criterion is given by

$$\delta_N = \sqrt{\frac{\sum_{j=N+1}^{n_s} \theta_j}{\sum_{i=1}^{n_s} \theta_i}} = \sqrt{\frac{\sum_{j=N+1}^{n_s} \theta_j}{||\underline{\underline{S}}||_F^2}} = \sqrt{||\underline{\underline{S}}||_F^2 - \sum_{j=1}^N \theta_j} \overset{!}{\leq} \varepsilon,$$

(14)

where $\varepsilon > 0$ is a given tolerance denoting the admissible approximation error. Then, the reduced basis is computed via

$$\underline{\underline{B}} = \underline{\underline{S}}\,\widetilde{\underline{\underline{V}}}\,\widetilde{\underline{\underline{\Theta}}}^{-\frac{1}{2}}$$

(15)

after truncation of the eigenvalue and eigenvector matrices to reduced dimension $N$ represented by $\widetilde{\underline{\underline{\Theta}}} \in \mathbb{R}^{N \times N}$ and $\widetilde{\underline{\underline{V}}} \in \mathbb{R}^{n \times N}$, respectively. The sorting of the eigenvalues with their corresponding eigenvectors leads to the property that the least recurrent information given in $\underline{\underline{S}}$ is omitted. Hence, the first eigenmode in $\underline{\underline{B}}$ has the most dominant pattern, the second eigenmode the second most, etc. The properties of the reduced basis computed with the snapshot correlation matrix remain the same as for the singular value decomposition (SVD) introduced below.

The SVD [38] of the snapshot matrix is given by

$$\underline{\underline{S}} = \underline{\underline{U}}\,\underline{\underline{\Sigma}}\,\underline{\underline{W}}^\mathsf{T}$$

(16)

with the following properties (asserting $n_s \geq n$)

$$\underline{\underline{U}} \in \mathbb{R}^{n \times n_s} : \underline{\underline{U}}^\mathsf{T} \underline{\underline{U}} = \underline{\underline{I}}, \qquad \underline{\underline{W}} \in \mathbb{R}^{n_s \times n_s} : \underline{\underline{W}}^\mathsf{T} \underline{\underline{W}} = \underline{\underline{I}}, \qquad \underline{\underline{\Sigma}} \in \mathbb{R}^{n_s \times n_s} : \underline{\underline{\Sigma}} = \operatorname{diag}(\sigma_i) \qquad (17)$$

and the sorted non-negative singular values $\sigma_i$ such that $\sigma_1 \geq \sigma_2 \geq \cdots \geq \sigma_{n_s} \geq 0$. The criterion for determining the reduced dimension $N$ matching Equation (14) takes the form

$$\delta_N = \sqrt{\frac{\sum_{j=N+1}^{n_s} \sigma_j^2}{\|\underline{\underline{S}}\|_F^2}} = \sqrt{\frac{\sum_{j=N+1}^{n_s} \sigma_j^2}{\sum_{i=1}^{n_s} \sigma_i^2}} = \sqrt{\|\underline{\underline{S}}\|_F^2 - \sum_{j=1}^{N} \sigma_j^2} \overset{!}{\leq} \varepsilon. \qquad (18)$$

Then the reduced basis is given by truncation of the columns of $\underline{\underline{U}}$ yielding $\widetilde{\underline{\underline{U}}} \in \mathbb{R}^{n \times N}$

$$\underline{\underline{B}} = \widetilde{\underline{\underline{U}}}. \qquad (19)$$

More specifically, the left subspace associated with the leading singular values represents the RB. Both introduced methods yield the exact same result for the same snapshot matrix $\underline{\underline{S}}$.

## 2.3. Incremental Generation of the Reduced Basis $\underline{\underline{B}}$

The RB is deployed in order to compress the information contained in $n_s$ snapshots into an $N$-dimensional set of eigenmodes stored in the columns of $\underline{\underline{B}} \in \mathbb{R}^{n \times N}$, where $N \ll n_s$ is asserted. Since the RB is computed with the snapshot matrix alone, the information contained in $\underline{\underline{S}}$ needs to contain data representing the relevant microstructure range, i.e., covering the parameter range used in the generation of the synthetic materials, in order for $\underline{\underline{B}}$ to be representative for the problem under consideration.

In the case of bi-phasic microstructural images containing $n$ pixels, a ludicrous amount of $2^n$ states could theoretically be considered when allowing for fully arbitrary microstructures. When limiting attention to certain microstructure classes, then less information is needed. Still, thousands of snapshots are usually required, at least. In the following, attention is limited to synthetic materials generated using random sequential adsorption of morphological prototypes with variable size, orientation, aspect ratio, overlap and randomized phase volume fraction. Due to the high variability of such microstructures (see, e.g., Figure 1), a large number of snapshots exceeding available memory would be needed, i.e., a monolithic snapshot matrix $\underline{\underline{S}}$ is not at hand in practice. While attention is limited to two-dimensional model problems in this study, the problem aggravates considerably for three-dimensional images which imply technical challenges of various sort (storage, processing time, data management, etc.).

In order to be able to generate a rich RB accounting for largely varying microstructural classes, the incremental basis generation represents a core concept within the present work. It enables the RB generation based on a sequence of input snapshots but without the need to store previously considered data except for the current RB. Three different methods are proposed, two of which rely on approximations of the snapshot correlation matrix $\underline{\underline{C}}_S$, and one of which relies on the SVD of an approximate snapshot matrix. The general incremental scheme depicted in Figure 2 remains the same for all the procedures, i.e., the only difference is found during the step labeled 'adjust'.

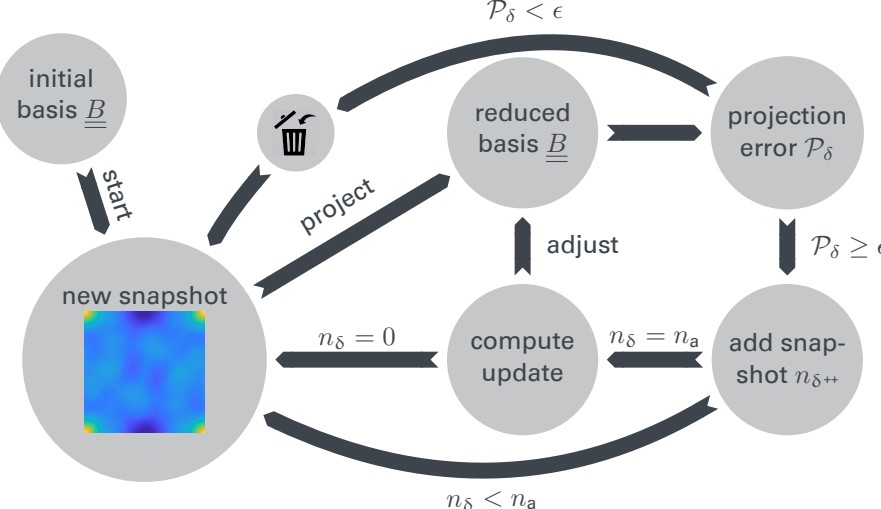

**Figure 2.** Graphical overview of the incremental update of the reduced basis.

The algorithm is initialized by a small sized set of initial snapshots of the shifted and scaled 2-point correlation function (cf. Equation (10) in Section 2.2). Further, the algorithmic variables $n_\delta = 0$ and $\underline{\underline{\Delta S}} = \varnothing$ are set. The initial RB is computed classically using either the correlation matrix or the SVD (see previous section for details). After computation of the RB, the snapshots are stored neither in memory nor on a hard drive. The algorithm then takes input snapshots in the order of appearance, i.e., the data gets abandoned. For each newly generated snapshot $\underline{s}_i$ the relative projection error with respect to the current RB is computed

$$\mathcal{P}_\delta = \frac{||\underline{s}_i - \underline{\underline{B}}\,\underline{\underline{B}}^{\mathsf{T}}\,\underline{s}_i||_F}{||\underline{s}_i||_F} . \tag{20}$$

If $\mathcal{P}_\delta$ is greater than the tolerance $\varepsilon > 0$ the snapshot is considered as inappropriately represented by the existing RB. Consequently, $\underline{s}_i$ is appended to a buffer $\underline{\underline{\Delta S}}$ containing candidates for the next basis enrichment and the counter $n_\delta$ is incremented. Once the buffer contains a critical number of $n_a$ elements the actual enrichment is triggered and the buffer is emptied thereafter. Thereby the computational overhead is reduced. The three different update procedures are described later on in detail. The procedure is continued until $n_c > 0$ consecutive snapshots were found to be approximated up to the relative tolerance $\varepsilon$. Then the basis is considered as converged for the microstructure class under consideration.

In the following three methods for the update procedure are described. Formally, the update of an existing basis $\underline{\underline{B}}$ with a block of snapshots contained in the buffer $\underline{\underline{\Delta S}}$ is sought-after. The new basis is required to remain orthonormal.

### 2.3.1. Method A: Append Eigenmodes to $\underline{\underline{B}}$

A trivial enrichment strategy is given in terms of appending new modes to the existing basis while preserving orthonormality of the basis. Therefore, the projection of $\underline{\underline{\Delta S}}$ onto the existing RB is subtracted in a first step

$$\underline{\underline{\Delta \widehat{S}}} = \underline{\underline{\Delta S}} - \underline{\underline{B}}\,\underline{\underline{B}}^{\mathsf{T}}\underline{\underline{\Delta S}}. \tag{21}$$

It is readily seen that $\underline{\underline{\Delta \widehat{S}}}$ is orthogonal to $\underline{\underline{B}}$. Then the correlation matrix of the additional data and its eigen-decomposition are computed according to

$$\underline{\underline{\Delta C}} = \underline{\underline{\Delta \widehat{S}}}^{\mathsf{T}}\,\underline{\underline{\Delta \widehat{S}}} = \underline{\underline{V}}\,\underline{\underline{\Theta}}\,\underline{\underline{V}}^{\mathsf{T}}. \tag{22}$$

Eventually, the enrichment is given through the truncated matrices $\widetilde{\underline{V}}$ and $\widetilde{\underline{\underline{\Theta}}}$

$$\underline{\underline{\Delta B}} = \underline{\underline{\Delta \widehat{S}}}\, \widetilde{\underline{V}}\, \widetilde{\underline{\underline{\Theta}}}^{-\frac{1}{2}}. \tag{23}$$

The new basis is then obtained by appending the newly computed modes $\underline{\underline{\Delta B}}$

$$\underline{\underline{B}} \leftarrow \begin{bmatrix} \underline{\underline{B}} & \underline{\underline{\Delta B}} \end{bmatrix}. \tag{24}$$

Method A simply adds modes generated from the projection residual $\underline{\underline{\Delta \widehat{S}}}$ in a decoupled way, i.e., the existing basis is not modified. In order to compute the basis update, only the existing RB $\underline{\underline{B}}$ and the temporarily stored snapshots $\underline{\underline{\Delta S}}$ are required.

Remarks on Method A

**A.1** The truncation parameter $\delta_N$ must be chosen carefully such that

$$\frac{\|\underline{\underline{\Delta \widehat{S}}} - \underline{\underline{\Delta B}}\, \underline{\underline{\Delta B}}^{\mathsf{T}} \underline{\underline{\Delta \widehat{S}}}\|_F}{\|\underline{\underline{\Delta S}}\|_F} \leq \delta_N. \tag{25}$$

In particular, the normalization with respect to the original data prior to projection onto the existing RB must be taken.

**A.2** By appending orthonormal modes to the existing basis it is a priori guaranteed that the accuracy of previously considered snapshots cannot worsen, i.e., an upper bound for the relative projection error of all snapshots considered until termination of the algorithm is given by the truncation parameter $\delta_N$ and $n_a$:

$$\max \frac{|\underline{s}_i - \underline{\underline{B}}\, \underline{\underline{B}}^{\mathsf{T}} \underline{s}_i|}{|\underline{s}_i|} \leq \sqrt{n_a}\, \delta_N. \tag{26}$$

This estimate is, however, overly pessimistic and it must be noted that the enrichment will guarantee a drop in the residual for all snapshots contained in $\underline{\underline{\Delta S}}$

### 2.3.2. Method B: Approximate Reconstruction of the Snapshot Correlation Matrix

This update scheme is based on an approximation of the new correlation matrix

$$\underline{\underline{C}} = \begin{bmatrix} \underline{\underline{S}}^{\mathsf{T}} \underline{\underline{S}} & \underline{\underline{S}}^{\mathsf{T}} \underline{\underline{\Delta S}} \\ \underline{\underline{\Delta S}}^{\mathsf{T}} \underline{\underline{S}} & \underline{\underline{\Delta S}}^{\mathsf{T}} \underline{\underline{\Delta S}} \end{bmatrix} = \begin{bmatrix} \underline{\underline{C}}_0 & \underline{\underline{S}}^{\mathsf{T}} \underline{\underline{\Delta S}} \\ \underline{\underline{\Delta S}}^{\mathsf{T}} \underline{\underline{S}} & \underline{\underline{\Delta S}}^{\mathsf{T}} \underline{\underline{\Delta S}} \end{bmatrix}. \tag{27}$$

Here $\underline{\underline{S}}$ denotes all snapshots considered in the RB so far and $\underline{\underline{\Delta S}}$ contains the candidate snapshots. However, the previously used snapshots formally written as $\underline{\underline{S}}$ are no longer available since they can not be stored due to storage limitations. Using the previously computed matrices $\underline{\underline{B}}, \widetilde{\underline{V}}, \widetilde{\underline{\underline{\Theta}}}$ the following approximations are available

$$\underline{\underline{S}}^{\mathsf{T}} \underline{\underline{S}} = \underline{\underline{C}}_0 \approx \widetilde{\underline{\underline{C}}}_0 = \widetilde{\underline{V}}\, \widetilde{\underline{\underline{\Theta}}}\, \widetilde{\underline{V}}^{\mathsf{T}}, \qquad \underline{\underline{B}} = \underline{\underline{S}}\, \widetilde{\underline{V}}\, \widetilde{\underline{\underline{\Theta}}}^{-\frac{1}{2}}, \qquad \underline{\underline{S}} \approx \underline{\underline{B}}\, \underline{\underline{B}}^{\mathsf{T}} \underline{\underline{S}}, \tag{28}$$

where the accuracy of the approximation is governed by the truncation threshold $\delta_N$. Using these approximations and using intrinsic properties of the spectral decomposition, the snapshot matrix $\underline{\underline{S}}$ up to the last basis adjustment is approximated by

$$\underline{\underline{S}} \approx \underline{\underline{B}}\, \widetilde{\underline{\underline{\Theta}}}^{\frac{1}{2}}\, \widetilde{\underline{V}}^{\mathsf{T}}. \tag{29}$$

Note that $\underline{\underline{B}} \in \mathbb{R}^{n \times N}$ is stored anyway, $\underline{\underline{\widetilde{\Theta}}} \in \mathbb{R}^{N \times N}$ is diagonal and $\underline{\underline{\widetilde{V}}} \in \mathbb{R}^{n_S \times N}$ is of manageable size (here $n_S \ll n$ is the number of snapshots with $\mathcal{P}_\delta \geq \epsilon$ considered in the basis generation up to now). The snapshot correlation matrix $\underline{\underline{C}}$ that considers the additional snapshots can be approximated as

$$
\underline{\underline{C}} \approx
\begin{bmatrix}
\underline{\underline{\widetilde{C}}}_0 & \underline{\underline{\widetilde{V}}}\,\underline{\underline{\widetilde{\Theta}}}^{\frac{1}{2}}\,\underline{\underline{B}}^{\mathsf{T}}\,\underline{\underline{\Delta S}} \\
\underline{\underline{\Delta S}}^{\mathsf{T}}\,\underline{\underline{B}}\,\underline{\underline{\widetilde{\Theta}}}^{\frac{1}{2}}\,\underline{\underline{\widetilde{V}}}^{\mathsf{T}} & \underline{\underline{\Delta S}}^{\mathsf{T}}\,\underline{\underline{\Delta S}}
\end{bmatrix}
=
\underbrace{
\begin{bmatrix}
\underline{\underline{\widetilde{V}}} & \underline{\underline{0}} \\
\underline{\underline{0}} & \underline{\underline{I}}
\end{bmatrix}
}_{\underline{\underline{V}}_*}
\underbrace{
\begin{bmatrix}
\underline{\underline{\widetilde{\Theta}}} & \underline{\underline{\widetilde{\Theta}}}^{\frac{1}{2}}\,\underline{\underline{B}}^{\mathsf{T}}\,\underline{\underline{\Delta S}} \\
\text{sym.} & \underline{\underline{\Delta S}}^{\mathsf{T}}\,\underline{\underline{\Delta S}}
\end{bmatrix}
}_{\underline{\underline{C}}_1}
\underbrace{
\begin{bmatrix}
\underline{\underline{\widetilde{V}}}^{\mathsf{T}} & \underline{\underline{0}} \\
\underline{\underline{0}} & \underline{\underline{I}}
\end{bmatrix}
}_{\underline{\underline{V}}_*^{\mathsf{T}}}.
\tag{30}
$$

In order to compute the updated basis, the inexpensive eigenvalue decomposition of $\underline{\underline{C}}_1 \in \mathbb{R}^{(N+n_a) \times (N+n_a)}$ is computed

$$
\underline{\underline{C}}_1 = \underline{\underline{V}}_1\,\underline{\underline{\Theta}}_1\,\underline{\underline{V}}_1^{\mathsf{T}}.
\tag{31}
$$

Analogously to the previous RB computation in Equation (15), the adjusted and enriched basis is computed by

$$
\underline{\underline{B}} = \begin{bmatrix} \underline{\underline{S}} & \underline{\underline{\Delta S}} \end{bmatrix} \underline{\underline{\widetilde{V}}}\,\underline{\underline{\widetilde{\Theta}}}^{-\frac{1}{2}} \approx \begin{bmatrix} \underline{\underline{B}}\,\underline{\underline{\widetilde{\Theta}}}^{\frac{1}{2}}\,\underline{\underline{\widetilde{V}}}^{\mathsf{T}} & \underline{\underline{\Delta S}} \end{bmatrix} \underbrace{\underline{\underline{V}}_*\,\underline{\underline{\widetilde{V}}}_1}_{\underline{\underline{\widetilde{W}}} \in \mathbb{R}^{(n_S+n_a) \times N}}\,\underline{\underline{\Theta}}_1^{-\frac{1}{2}}.
\tag{32}
$$

To update the RB the truncated eigenvector matrix $(\underline{\underline{B}}, \underline{\underline{\widetilde{V}}} \leftarrow \underline{\underline{\widetilde{W}}} \in \mathbb{R}^{(n_S+n_a) \times N})$ need to be stored as well as the diagonal eigenvalue matrix $\underline{\underline{\widetilde{\Theta}}}$.

Remarks on Method B

**B.1**　The existing RB is not preserved but it is updated using the newly available information. Thereby, the accuracy of the RB for the approximation of the previous snapshots is not guaranteed a priori. However, numerical experiments have shown no increase in the approximation errors of previously well-approximated snapshots.

**B.2**　In contrast to Method A the dimension of the RB can remain constant, i.e., a mere adjustment of existing modes is possible. The average number of added modes per enrichment is well below that of Method A.

**B.3**　The additional storage requirements are tolerable and the additional computations are of low algorithmic complexity. In particular, the correlation matrix $\underline{\underline{C}}_1$ consists of a diagonal block complemented by a dense rectangular block, rendering the eigenvalue decomposition more affordable.

### 2.3.3. Method C: Incremental SVD

Method C is closely related to Method B. However, instead of building on the use of the correlation matrix, it relies on the use of an updated SVD, i.e., an approximate truncated SVD is sought after

$$
\operatorname{trunc} \operatorname{svd} \left( \begin{bmatrix} \underline{\underline{S}} & \underline{\underline{\Delta S}} \end{bmatrix} \right) \approx \underline{\underline{B}}\,\underline{\underline{\Sigma}}\,\underline{\underline{W}}^{\mathsf{T}}.
\tag{33}
$$

Since the original snapshot matrix $\underline{\underline{S}}$ can not be stored, only an approximation of the actual truncated SVD in (33) can be computed. Methods to compute an incremental SVD were, e.g., introduced in [39,40], with the latter referring to Brand's incremental algorithm [41] which is used in the present study with minor modifications. With the previously computed basis $\underline{\underline{B}}$ at hand, the approximation of $\underline{\underline{S}}$ is known

$$
\underline{\underline{S}} \approx \underline{\underline{B}}\,\underline{\underline{\Sigma}}\,\underline{\underline{W}}^{\mathsf{T}}.
\tag{34}
$$

First, the projection residual $\underline{\underline{\Delta \widehat{S}}}$ of the enrichment snapshots $\underline{\underline{\Delta S}}$ and its SVD

$$
\underline{\underline{\Delta \widehat{S}}} = \underline{\underline{\Delta S}} - \underline{\underline{B}}\,\underline{\underline{B}}^{\mathsf{T}}\,\underline{\underline{\Delta S}} \quad = \underline{\underline{U}}_S\,\underline{\underline{\Sigma}}_S\,\underline{\underline{W}}_S^{\mathsf{T}},
\tag{35}
$$

are computed. By using the truncated SVD to approximate the previous snapshots cf. Equation (34) and accounting for the newly added snapshots via Equation (35), the new snapshot matrix including the candidate snapshots can be approximated by

$$
\begin{bmatrix} \underline{S} & \underline{\Delta S} \end{bmatrix} \approx \begin{bmatrix} \underline{B}\,\underline{\Sigma}\,\underline{W}^\mathsf{T} & \underline{\Delta S} \end{bmatrix} = \begin{bmatrix} \underline{B} & \underline{U}_S \end{bmatrix} \underbrace{\begin{bmatrix} \underline{\Sigma} & \underline{B}^\mathsf{T}\underline{\Delta S} \\ \underline{0} & \underline{\Sigma}_S\,\underline{W}_S^\mathsf{T} \end{bmatrix}}_{\underline{\underline{\Gamma}}} \begin{bmatrix} \underline{W} & \underline{0} \\ \underline{0} & \underline{I} \end{bmatrix}^\mathsf{T}. \tag{36}
$$

The matrix $\underline{\underline{\Gamma}}$ consists of a $N \times N$ diagonal block and a rectangular matrix of size $(N + n_a) \times n_a$. Due to this sparsity pattern, the SVD $\underline{\underline{\Gamma}} = \underline{\underline{U}}_\Gamma\,\underline{\underline{\Sigma}}_\Gamma\,\underline{\underline{W}}_\Gamma^\mathsf{T} \in \mathbb{R}^{(N+n_a)\times(N+n_a)}$ is inexpensive to compute. It allows to rewrite Equation (36) as

$$
\begin{bmatrix} \underline{S} & \underline{\Delta S} \end{bmatrix} \approx \Big( \underbrace{\begin{bmatrix} \underline{B} & \underline{U}_S \end{bmatrix} \underline{U}_\Gamma}_{\underline{\underline{U}}_*} \Big) \underbrace{\underline{\Sigma}_\Gamma}_{\underline{\underline{\Sigma}}_*} \Big( \underbrace{\begin{bmatrix} \underline{W} & \underline{0} \\ \underline{0} & \underline{I} \end{bmatrix} \underline{W}_\Gamma}_{\underline{\underline{W}}_*} \Big)^\mathsf{T}. \tag{37}
$$

It is easily shown that the matrices $\underline{\underline{U}}_*$ and $\underline{\underline{W}}_*$ are column-orthogonal and that $\underline{\underline{\Sigma}}_*$ is diagonal and non-negative. Therefore, the three matrices constitute an approximate SVD of the enlarged snapshot matrix at low computational expense. This implies the following updates after the enrichment step

$$
\underline{\underline{B}} \leftarrow \begin{bmatrix} \underline{B} & \underline{U}_S \end{bmatrix} \underline{U}_\Gamma \qquad\qquad \underline{\underline{\Sigma}} \leftarrow \widetilde{\underline{\Sigma}}_\Gamma \qquad\qquad \underline{\underline{W}} \leftarrow \begin{bmatrix} \underline{W} & \underline{0} \\ \underline{0} & \underline{I} \end{bmatrix} \underline{W}_\Gamma \tag{38}
$$

after truncation of $\underline{\underline{B}}$, where the truncation criteria needs to ensure that $\underline{\underline{B}}$ does not decrease in size. To compute the enrichment of the RB, $\underline{\underline{B}} \in \mathbb{R}^{n\times N}$ and the sparse singular values $\underline{\underline{\Sigma}} \in \mathbb{R}^{N\times N}$ after truncation need to be stored.

Remarks on Method C

**C.1**　As highlighted for Method B (see remark **B.1**), the existing RB is not preserved but adjusted by considering the newly added information. A priori guarantees regarding the subset approximation accuracy can not be made, i.e., the approximation error of the previous snapshots $\underline{\underline{S}}$ could theoretically worsen. However, our numerical experiments did not exhibit such behavior at any point.

**C.2**　In contrast to Method A the dimension of the RB can remain constant, i.e., a mere adjustment of existing modes is possible. The average number of added modes per enrichment is well below that of Method A.

**C.3**　Each update step in (38) is computed separately and, consequently, storing $\underline{\underline{W}}$ is not required since only the RB $\underline{\underline{B}}$ is of interest.

**C.4**　The diagonal matrix $\underline{\underline{\Sigma}}$ has low storage requirements corresponding to that of a vector in $\mathbb{R}^N$.

### 3. Supervised Learning Using Feed Forward Neural Network

During the supervised learning phase, the machine is provided with data sets consisting of inputs and the related outputs: We aim at learning an unknown function relating inputs (here: Image data compressed into a low dimensional feature vector) to outputs (here: Effective thermal conductivity tensors) without or with limited prior knowledge of the structure of this function. Artificial Neural Networks (ANN) are a powerful machine learning tool which has gained wide popularity in the recent years due to the surge in computational power [27,42] and the availability of easy to use software packages (as a frontend in Python: Keras, Pytorch, TensorFlow or as graphical user interfaces Neuraldesigner amongst many others).

The functionality of the ANN is inspired by the (human) brain, propagating a signal (input) through multiple neurons where it is lastly transformed into an action (output). Various types of neural networks have been invented, e.g., feedforward, recurrent or convolutional networks, being applicable to almost any field of interest [43–46].

In the present study a regression model from the input, i.e., the feature vector $\underline{\xi}$ which is derived with the converged basis $\underline{B}$, to the output, i.e., the effective heat conduction tensor $\underline{\bar{\kappa}}$, is deployed with a dense feedforward ANN.

In a dense feedforward ANN (Figure 3) a signal is propagated through the hidden layers where every output of the previous layer $\underline{a}^{l-1}$ affects the activation $\underline{z}^l$ of the current layer $l$ ($l = 1, \ldots, L+1$). The activation of each layer gets wrapped into an activation function $f$ where the output of each neuron in the layers is computed, i.e., $\underline{a}^l = f(\underline{z}^l)$. Note that matrix/vector notation is used, where each entry in the vectors denotes one neuron in the respective layer.

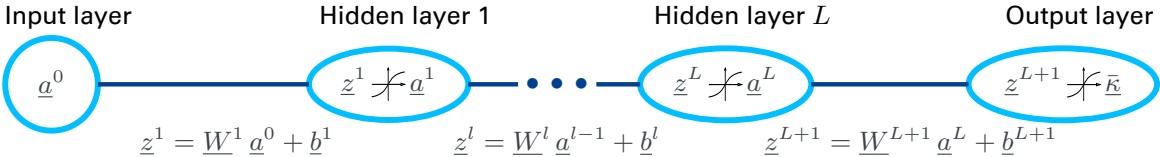

**Figure 3.** The basic functionality of a dense feedforward neural network is depicted in simplified form.

The basic learning algorithm/optimizer usually employed for a feedforward ANN is the back propagation algorithm [47] and modifications thereof. The learning of the network consists in the numerical identification of the unknown weights $\underline{W}^l$ and biases $\underline{b}^l$ minimizing a given cost function, where a random initialization defines the initial guess for all parameters. The cost function gives an indication of the quality of the ANN prediction. The gradient back propagation computes suitable corrections for the parameters of the ANN by evaluating the gradients of the cost function to the weights.

The learning itself is an iterative procedure in which the training data is cycled multiple times through the ANN (one run called an 'epoch'). In each epoch the internal parameters are updated with the aim of improving the mapping relating input and output data, aiming at reduction of the cost function. The optimization problem itself is (usually) high-dimensional. In most situations it is not well-posed and local minima and maxima can hinder convergence to the global minimum. Therefore, multiple random instantations of the network parameters are usually required to assure that a good set of parameters is found, even if the network layout remains unaltered.

The training requires a substantial input data set as input-output tuples in order to allow for robust and accurate predictions.

It is important to note that the (repeated) training of the ANN usually results in a parameter set that is able to approximate the training data with high accuracy under the given meta-parameters describing the network architecture (number of layers, number of neurons per layer, type of activation function). However, the approximation quality of the ANN may be different for query points not contained in the training set. Thus, it is important to validate the generality of the discovered surrogate

for the underlying problem setting. Therefore, an additional validation data set is introduced, where only the evaluation of the cost function is tracked over the epochs. Generally, when overfitting occurs (overfitting relates to the fact that a subset of the data is nicely matched but small variations in the inputs can lead to substantial loss in accuracy, similar to oscillating higher-order polynomial interpolation functions), the errors for the validation set increase whereas the errors of the training set decrease. The training should be halted if such a scenario is detected.

Since the choice of activation function as well as the number of hidden layers and the number of neurons within the individual layers are arbitrary (describing the ANN architecture), these meta-parameters should be tailored specifically for the desired mapping. Finding the best neural network architecture is not straight-forward and usually relies on intuition, experience and a substantial amount of numerical experiments. As mentioned earlier, the identification of a well-suited ANN requires various random realizations (corresponding to different initial biases and weights) for each ANN architecture under consideration. The optimum is then found as the best ANN over all realizations over all tested architectures.

In the present study the ANN training is performed using TensorFlow in Python [48]. TensorFlow is an open source project by the Google team, providing highly efficient algorithms for ANN implementation. The ADAM [49] optimizer, which is a modification of the gradient back propagation, has been deployed for the learning.

## 4. Results

### 4.1. Generation of Synthetic Microstructures

All of the used synthetic microstructures have been generated by a random sequential adsortion algorithm with some examples shown in Figure 1. Two morphological prototypes were used: spheres and rectangles. The deployed microstructure generation algorithm ensures a broad variability in the resulting microstructure geometry. Indeed, any bi phasic microstructure image can be considered. The parameters used to instantiate the generation of a new microstructure were modeled as uniformly distributed variables:

**M.1**　　The phase volume fraction $f_b$ of the inclusions (0.2–0.8);
**M.2**　　The size of each inclusion (0.0–1.0);
**M.3**　　For rectangles: The orientation ($0$–$\pi$) and the aspect ratio (1.0–10.0);
**M.4**　　The admissible relative overlap $\varrho$ for each inclusion (0.0–1.0).

For $\varrho = 0$ and the spherical inclusion, a boolean model of hard spheres is obtained. Setting $\varrho = 1$ induces a boolean model without placement restrictions, i.e., new inclusions can be placed independent of the existing ones. The generated microstructures were stored as images with resolution $400 \times 400$. After the generation of the RVE, the 2-point spatial correlation function was computed for the RVE. This was then shifted and scaled, see Equation (10) in Section 2.2, and used as a snapshot $\underline{s}_i$ for the identification of the reduced basis.

Additionally, a smaller random set of RVEs used for the supervised learning phase was simulated using the recent Fourier-based solver FANS [3] in order to compute the effective heat conduction tensor $\underline{\underline{\bar{\kappa}}}$. The heat conductivity of the matrix and of the inclusion phase are prescribed as

$$\kappa_a = 1.0 \Big[ \frac{W}{m \cdot K} \Big], \qquad\qquad \kappa_b = \frac{\kappa_a}{R} \Big[ \frac{W}{m \cdot K} \Big]. \qquad\qquad (39)$$

Here $R > 0$ denotes the material contrast. In the present study, $R = 5$ was considered, i.e., the matrix of the microstructure has a five times higher conductivity than the inclusions. These values can be seen as typical values for metal ceramic composites (Figure 4).

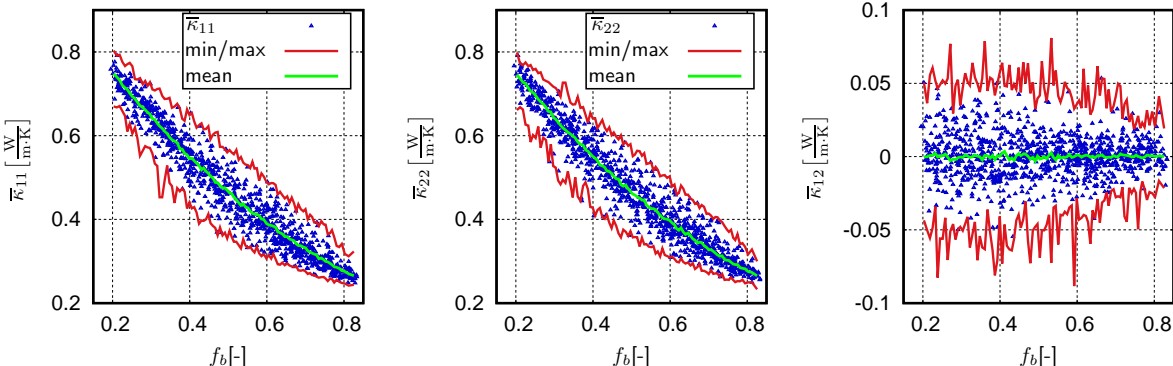

**Figure 4.** The range of each $\underline{\underline{\bar{\kappa}}}$ entry computed with 15,000 microstructures of the mixed set is shown. Only 1000 discrete values are shown in each plot.

An inverse phase contrast has exemplarily been studied, i.e., inclusions with $\kappa_b = 1\,\mathrm{W\,m^{-1}\,K^{-1}}$ and $\kappa_a = \frac{\kappa_b}{5}$ (corresponding to $R = \frac{1}{5}$) $R = 1/5$) has also been investigated. Qualitatively, the results for the inverse phase contrast did not show any new findings or qualitative differences. Therefore, the following results focus on $R = 5$, corresponding to rather insulating inclusions.

The symmetric tensor $\underline{\underline{\bar{\kappa}}}$ can be represented as a three-dimensional vector $\underline{\bar{\kappa}}$ using the normalized Voigt notation

$$\underline{\underline{\bar{\kappa}}} = \begin{bmatrix} \bar{\kappa}_{11} & \bar{\kappa}_{12} \\ \bar{\kappa}_{21} & \bar{\kappa}_{22} \end{bmatrix} \qquad \rightarrow \qquad \underline{\bar{\kappa}}_{\mathrm{V}} = \begin{bmatrix} \bar{\kappa}_{11} \\ \bar{\kappa}_{22} \\ \sqrt{2}\,\bar{\kappa}_{12} \end{bmatrix}. \tag{40}$$

For the supervised learning of the ANNs (see Section 3), multiple files each containing 1,500 data sets for different inclusion morphologies were generated (circle only; rectangle only; mixed; see following section). Each data set contains the image of the microstructure, the respective autocorrelation of the inclusion phase $c_2(\bullet; b, b)$ and the effective heat conductivity $\underline{\bar{\kappa}}_{\mathrm{V}}$.

## 4.2. Unsupervised Learning

First, the reduced basis is identified using the iterative procedure presented in Section 2.3. All three proposed methods were considered and for each of these, three different sets of microstructures were used as inputs: The first set of microstructures consisted of RVEs with only circular inclusions, the second set consisted of RVEs with only rectangular inclusions, and the third set was divided into equal parts, each part consisting of RVEs with either circular or rectangular inclusions (i.e., each structure contained exclusively one of the two morphological prototypes and the same number of realizations for each prototype was enforced), respectively. Each type of microstructure was processed using each of the three incremental RB schemes introduced in Section 2.3. Hence, a total of nine different trainings were conducted, each using different randomly generated snapshots.

For the iterative enrichment process, the initial RB was computed from 200 snapshots $\underline{\underline{S}}_0$. Thereafter, snapshots were randomly generated and processed by the enrichment algorithm sketched in Figure 2. The number of snapshots per enrichment step has been set to $n_{\mathrm{a}} = 75$ and the number of consecutive snapshots with $\mathcal{P}_\delta < \varepsilon$, used to indicate convergence, has been set to $n_{\mathrm{c}} = 100$. The relative projection tolerance $\varepsilon = 0.025$ was chosen. Note that this corresponds to the maximum value of the mean relative $\|\cdot\|_{L^2}$-error that is considered exact for the shifted and scaled snapshots. The actual accuracy in the reproduction of the 2PCF $c_2(\mathbf{r}; b, b)$ is significantly lower than this (results are given in Figure 7).

Key attributes for each of the nine trainings are provided in Table 1. There is an obvious discrepancy between Method A and the remaining methods in basically all outputs. While Method A claims the lowest computing times, it yields approximately twice the number of modes. However,

the number of snapshots needed is substantially lower which can be relevant if the generation of the synthetic microstructures is computationally involved.

**Table 1.** Data of the unsupervised learning (incremental reduced basis (RB) identification) for the nine considered scenarios; the parameters $\varepsilon = 0.025$, $n_c = 100$ and $n_a = 75$ were used. Some numbers are rounded for easier readability.

| Method | Final Basis Size | Snapshots with $\mathcal{P}_\delta \geq \varepsilon$ | Snapshots with $\mathcal{P}_\delta \leq \varepsilon$ | Enrichment Steps | Time [s] | Used Microstructures |
|---|---|---|---|---|---|---|
| A | 143 | 150 | 730 | 4 | 20 | |
| B | 80 | 400 | 2400 | 7 | 70 | |
| C | 96 | 800 | 7700 | 12 | 200 | |
| A | 596 | 670 | 4500 | 11 | 150 | |
| B | 294 | 2400 | 12,700 | 34 | 500 | |
| C | 312 | 2600 | 16,500 | 37 | 550 | |
| A | 464 | 560 | 2900 | 9 | 150 | |
| B | 274 | 2000 | 16,100 | 29 | 500 | |
| C | 244 | 1540 | 8000 | 22 | 280 | |

Note that methods B and C yield similar results, although for the rectangular and circular training Method C needed significantly more snapshots, Method B needed significantly more snapshots for the mixed training. The outliers between methods B and C in the number of snapshots needed are due to the randomness of the materials and the chosen convergence criterion. The resulting basis size of methods B and C indicate very similar results from these methods. Note that methods B and C yield identical results when operating on an identical sequence of microstructures used as inputs when leaving aside perturbations due to numerical truncation.

In addition, note that the computational effort for the relative projection error $\mathcal{P}_\delta$ grows linearly with the dimension of the RB, i.e., the faster offline time of Method A can quickly be compensated by the costly online procedure induced by the high dimension of the RB in comparison to the competing techniques.

To compare the accuracy of the resulting basis as well as during the training, the relative projection error $\mathcal{P}_\delta$ of the snapshots used for the original basis construction $\underline{\underline{S}}_0$ are plotted in Figure 5.

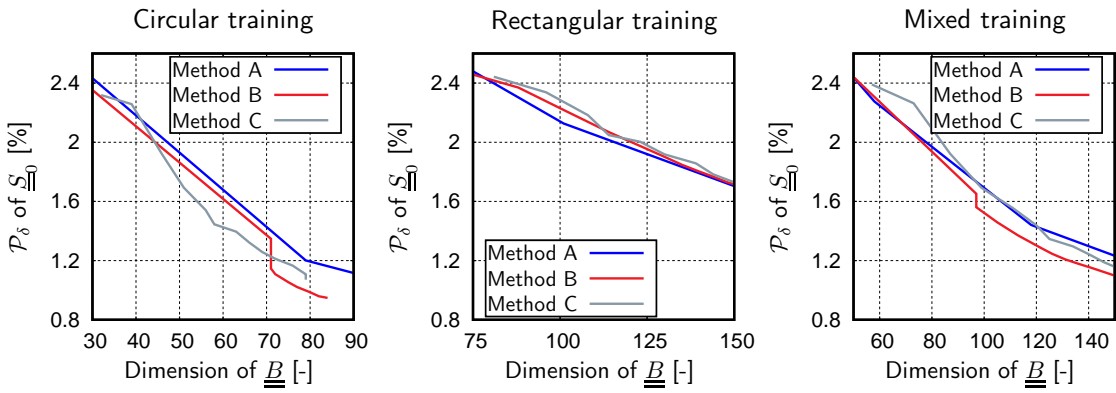

**Figure 5.** Development of the relative projection error $\mathcal{P}_\delta$ of the snapshots $\underline{\underline{S}}_0$ with respect to the current basis size $N$ over the enrichment.

While methods B and C do not, unlike Method A, a priori guarantee an improvement of the relative projection error of $\underline{S}_0$ over the enrichment, a strict downward trend is observed. The adjustment of already existing eigenmodes in methods B and C allow for an improvement of the relative projection error of $\underline{S}_0$ for a constant basis size.

Method B and C seem to outperform Method A in most cases; however, the basis of Method A achieves a lower projection error on convergence (not shown in the plot), but at the expense of a considerably larger dimension of the RB.

Since there seems to be an obvious correlation between resulting accuracy and the final basis size for the initial snapshots $\underline{S}_0$ (see Figure 5, Table 1), the general quality for arbitrary stochastic inputs must be investigated. In order to quantify the quality of the RB, the accuracy can be expressed in terms of the relative projection error of approximating additional, newly generated snapshot data $\underline{\underline{S}}$ as a function of the Method (A, B, C) and the number of modes $N \geq 1$ via

$$\mathcal{P}_\delta(N) = \sqrt{\frac{||\underline{\underline{S}} - \underline{\underline{B}}(:, 1:N)\,\underline{\underline{B}}^{\mathsf{T}}(:, 1:N)\,\underline{\underline{S}}||_F^2}{||\underline{\underline{S}}||_F^2}} \tag{41}$$

in Matlab notation.

This measure captures to what extend the first $N$ basis functions represent the 2PCF of the underlying microstructure class. In the current work sets of 1500 newly generated snapshots assure an unbiased validation, i.e., the data was used in neither of the three training procedures. The results are stated in Figure 6. Again, Method B and C yield similar results, achieving lower projection errors with fewer eigenmodes compared to Method A, i.e., the basis produced by Method A cannot catch up with its two competitors. On a side note, the rectangular inclusions apparently lead to significantly richer microstructure information which can be seen by direct comparison of the left to the middle plot in Figure 6. For methods B and C and for circular inclusions the relative error of 5% is reached for approximately 15 modes while rectangular inclusions require more than 60 modes to attain a similar accuracy. This is supported also by the rightmost plot determined from a sort of blend of the two microstructural types.

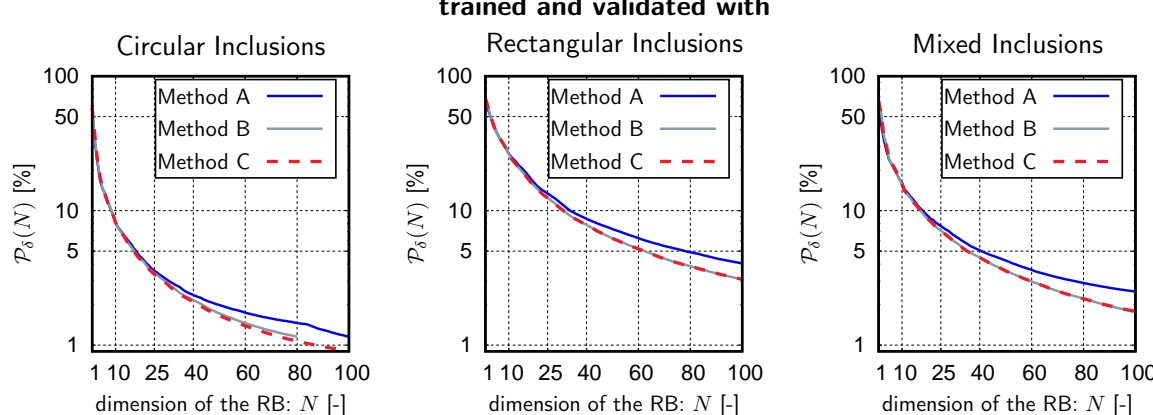

**Figure 6.** Relative projection error for three different microstructure classes as a function of the number of eigenmodes. The relative projection error is determined for a validation set of 1,500 newly generated microstructures for each class.

Since all of the previous error measures are given on the shifted snapshot according to Equation (10), the true relative projection error on the unshifted snapshot is also investigated as a function of the basis size. It describes the actual relative accuracy of the approximation of the 2PCF $c_2(\boldsymbol{r}; b, b)$ as a function of the basis size. The errors in the shifted data (Figure 7, left) and the corresponding reconstructed 2PCF (Figure 7, right) for five randomly selected snapshots show that

the actual relative error in the 2PCF reconstruction is below 5% for 10 reduced coefficients even for the challenging rectangular inclusion morphology, while the error in the shifted and scaled snapshots is on the order of 50%. This highlights the statement made earlier regarding the choice of $\varepsilon$ which is not directly the accepted mean error in the 2PCF, but only after application of the shift. The high discrepancy in the two relative projection errors is due to the fact that the shifted snapshots fluctuate closely around 0, i.e., the homogeneous part of the 2PCF is obviously of high relevance.

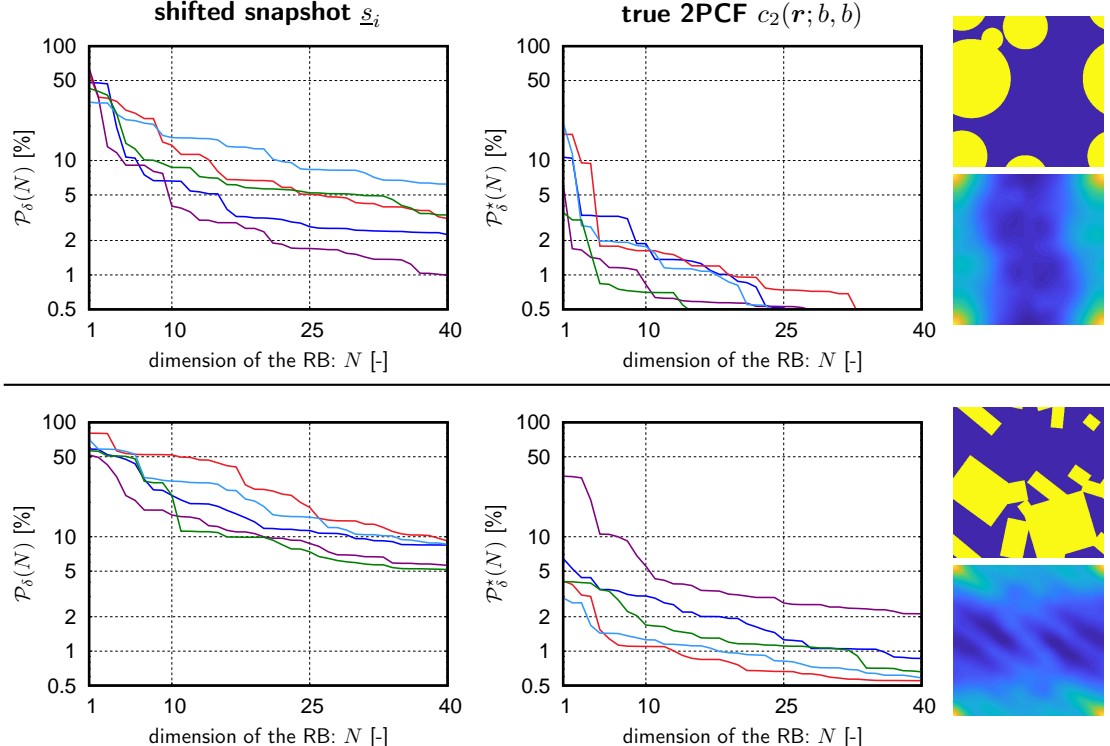

**Figure 7.** Using the RB of Method C, the relative projection error on the shifted snapshot $\mathcal{P}_\delta$ is given on the left for five random samples. For comparison the relative projection error of the reconstruction of the actual 2-point correlation function $\mathcal{P}_\delta^*$ is given on the right for the same five samples.

The development, i.e., the stabilization of the mode shapes over the enrichment steps, of a few selected eigenmodes is shown in Figure 8 using RVEs with circular inclusions for training of Method C. Similar results are expected for Method B, whereas for Method A the eigenmodes would remain unconditionally unchanged over the enrichment steps, i.e., a pure enlargement of the basis takes place. The faster stabilization of the leading eigenmodes indicates a quick stabilization of the lower order statistics of the microstructure ensemble, while the tracking of higher order fluctuations is more involved.

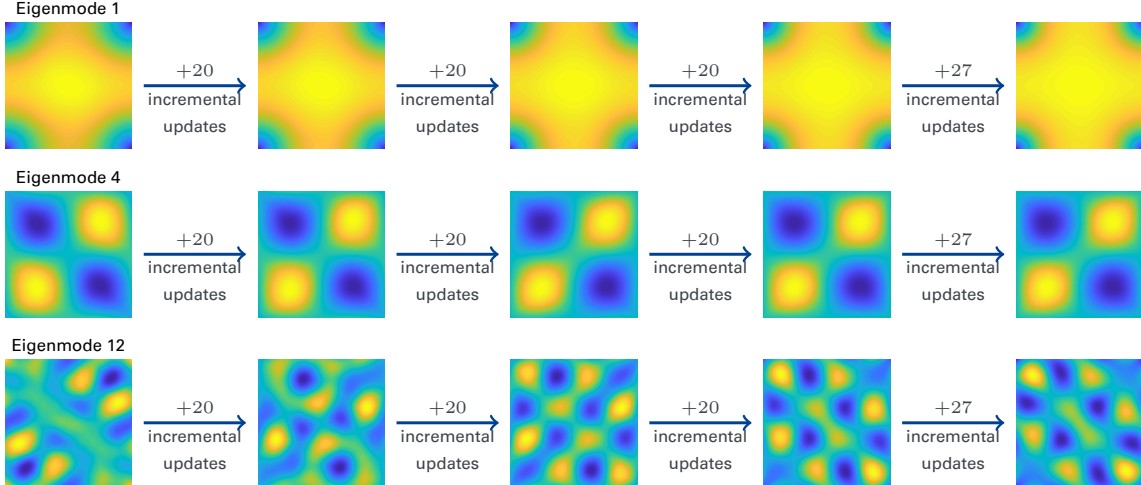

**Figure 8.** The development of a few selected eigenmodes over the enrichment are shown for the circular inclusion morphology. Note that these results are generated with $n_a = 15$ and $\varepsilon = 0.01$ using Method C. The procedure comprised a total of 87 basis enrichments/adjustment.

### 4.3. Supervised Learning

After the training of the RB, the input for the neural network, the feature vector $\underline{\xi}$ was derived using the 1- and 2-point spatial correlation functions of the $i$th RVE as

$$\underline{\xi}_i = \begin{bmatrix} f_{b,i} \\ \underline{\underline{B}}^{\mathsf{T}} \underline{s}_i \end{bmatrix} \quad \in \mathbb{R}^{(h+1)} . \tag{42}$$

The size of the feature vector is determined by the amount of reduced coefficients $1 \leq h \leq N$, i.e., the snapshot is projected onto the leading $h$ eigenmodes of $\underline{B}$.

Since the inputs and outputs have a highly varying magnitude, they need to be shifted such that they are equally representative. Therefore, each entry of the feature vector is separately shifted and scaled such that its distribution of all samples has zero mean and a standard deviation of one. The output is shifted combinedly such that the mean of $\underline{\bar{\kappa}}_V$ is $\underline{0}$. The transformed inputs and outputs are then given to the ANN for the training phase. Thus, the outputs of the ANN need to undergo an inverse scaling in order to yield the sought-after vector representation of the heat conduction tensor. These shifts and scalings need to be extracted from the available training data. Hence, every data set used for training purposes has its own parameters.

The training for the neural network has been conducted for all of the three microstructure classes, i.e., using only RVEs with circular inclusions, only RVEs with rectangular inclusions and lastly using RVEs with either circular or rectangular inclusions with equal number of realizations of each shape within the mixed set. In order to derive the feature vector, the converged basis of Method C has been used. Note that depending on the training set, either circular or rectangular or both inclusion shapes (for the mixed set) contributed to the RB.

In order to find a good overall ANN, the network architecture has been intensely studied: The accuracy of the prediction after the training has been evaluated with various sizes of the feature vector, different network layouts and for different activation functions (Figure 9).

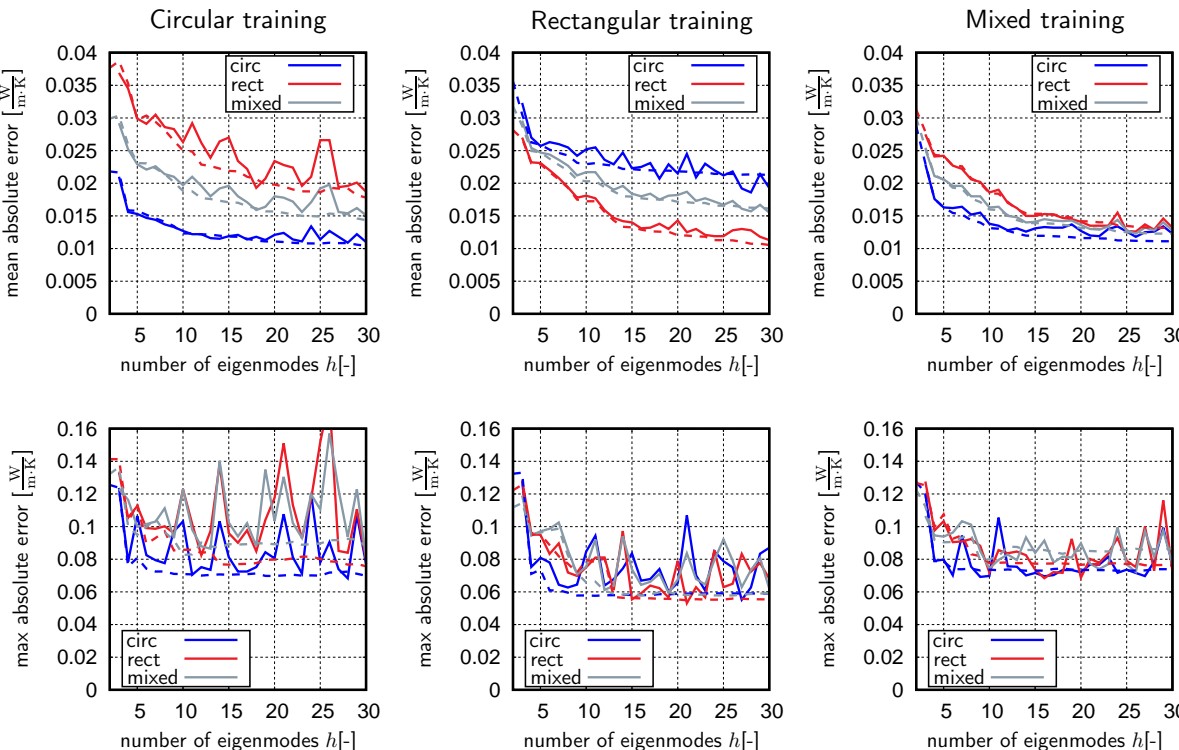

**Figure 9.** The given error measures over the test sets are shown for the Gaussian Process Model (GPM) (dashed lines) and the Artificial Neural Networks (ANN) (full lines) which achieved the lowest MSE (cost) on the validation set for each number of reduced coefficients and training type.

The training of the ANN was conducted with an early stop algorithm, stopping the training after 500 consecutive epochs of no improvement of the cost function with respect to the validation set. The learning rate of the ANN has been held constant during the training, being randomly initialized between 0.01 and 0.05. A network depth of up to 6 hidden layers and a network width of up to 100 hidden neurons have been considered and the number of neurons was chosen on a per layer basis. Recall that a vanilla dense feedforward ANN has been deployed. In order to find the best ANN architecture, 35 randomly initialized ANN trainings have been considered for each size of the feature vector. A total amount of 1500 samples have been considered for each ANN training. These were shuffled randomly and split into the training set ($n_t = 1000$) and the validation set ($n_v = 500$).

In the following, the error measurements used and the term of unbiased testing refers to the prediction of 7500 unseen data points for each of the three microstructure classes named 'test sets'.

The prediction error is given by the 2-norm, i.e.,

$$e^{\mathrm{P}} = ||\overline{\kappa}_{\mathrm{V}} - \overline{\kappa}_{\mathrm{V}}^{\mathrm{P}}||_2 \,, \tag{43}$$

with $\overline{\kappa}_{\mathrm{V}}^{\mathrm{P}}$ denoting the prediction of the regression model. The mean and maximum errors of the prediction error for all test sets are shown in Figure 9. For comparison of the regression model, we have deployed a Gaussian Process Model (GPM) [50], which reliably finds the global minimum of the optimization for the kernel regression. The ANN is given with full lines and the GPM model is given with dashed lines in Figure 9. Note that each ANN realization refers to a randomly initialized ANN architecture.

The GPM model seems to achieve slightly lower errors than the ANN, however, in the interest of computational speed the ANN regressor is preferred. Not only is the training significantly faster, the prediction times for the GPM highly depend on the size of the input vector, whereas the prediction times of the ANN mostly depend on the ANN architecture. More details are given in Section 5.

The spikes in Figure 9 regarding the maximum error, are explained by each depicted ANN having the lowest overall MSE of the validation set, which did not consider the maximum errors directly. Though, only a few outliers yielded a high prediction error, as can be seen in Figures 10 and 11.

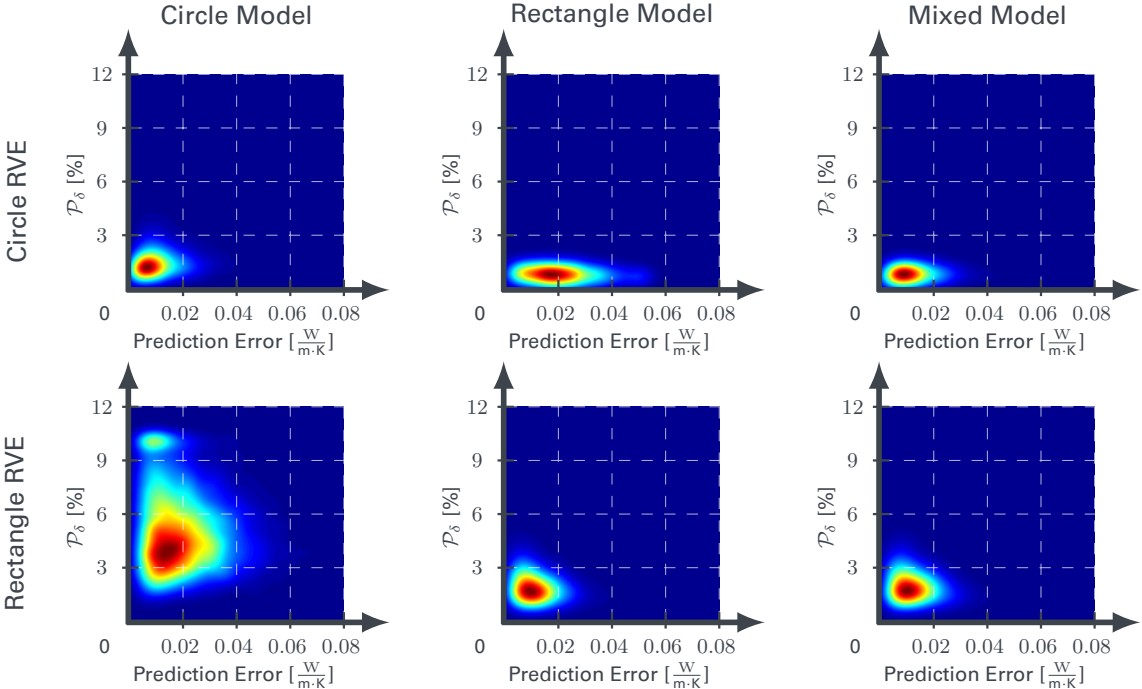

**Figure 10.** A density map of the projection error of the reduced basis compared to the prediction error of the ANN is given for all of the three training variants, for the prediction of the circle and rectangle test set, respectively. Each with one exemplary ANN (basis dimensions are 23, 25 and 25, respectively, from left to right).

Note that the ANN trained with rectangular RVE achieved lower maximum errors, whereas the circular RVE training achieved lower mean errors (Figure 9). A possible explanation is that rectangular inclusions allow for more complex geometries in the microstructure than perfectly, yet overlapping spherical inclusions. This possibly allows the RB as well as the ANN to better learn about microstructure geometries which, usually, lead to a high prediction error. The ANN trained with both microstructure classes manages to nicely capture both training advantages of the RVE classes and achieves a good mean accuracy as well as low maximum errors across the board.

The conductivity $\overline{\kappa}_{12}$ fluctuates mildly around zero for all inputs. In order to accurately capture this fluctuation, only the specific training and RB dimensions of four or higher ($h \geq 4$) are required cf. Figure 12. Albeit the values can be considered small in comparison to the $\overline{\kappa}_{11}$ and $\overline{\kappa}_{22}$ errors.

The overall downward trend of the prediction errors validate our approach, implying that a higher amount of reduced coefficients leads to more detailed information about the microstructure geometry, allowing for a better prediction of the regression model. However, the prediction errors do not seem to completely vanish, therefore the 2PCF alone does not suffice to perfectly describe the microstructure geometry.

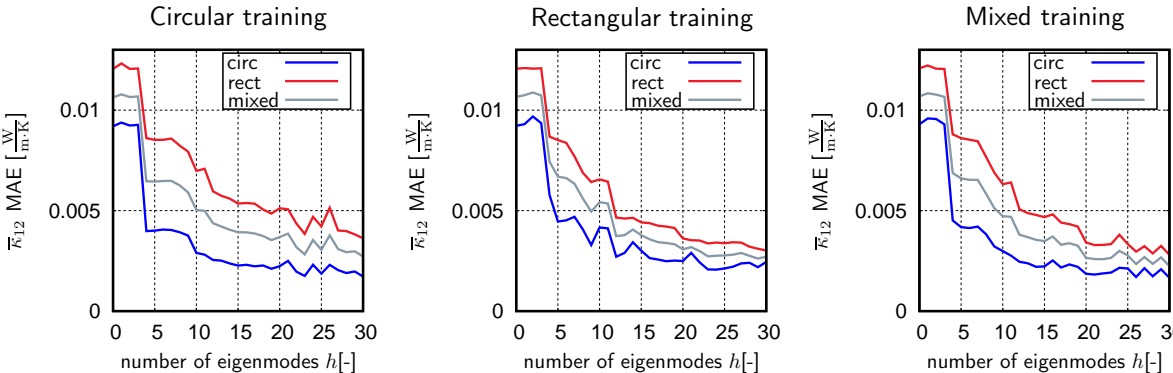

**Figure 11.** Results for the best of all tested ANN for the test sets. The graphs represent a probability distribution of the absolute error in each component of $\underline{\bar{\kappa}}$.

**Figure 12.** The mean absolute error (MAE) of $\bar{\kappa}_{12}$ is given for each of the training types and test sets.

To further study the accuracy of our surrogate model, which is divided into two processes namely the feature extraction with the RB and thereafter the prediction of the ANN, the error committed

in each step is examined in Figure 10. Intuitively, a high projection error of the reduced basis is expected to yield poor knowledge of the microstructure geometry and, as a consequence, lead to a high prediction error of the ANN. On the contrary, microstructures with the highest projection errors still allowed for accurate ANN predictions and the highest ANN prediction errors occurred for relatively small projection errors. The comparison of the RB relative projection error plotted against the GPM prediction error yielded very similar results. Note that the relatively high projection errors on the circle-trained RB are due to the fact that the basis is significantly smaller, leading to an overall higher projection error (Table 1). The relative projection errors have been measured on the shifted and scaled 2PCF which is a more pessimistic prediction than the actual 2PCF cf. the results shown in Figure 7.

An observation of the worst predictions for each ANN (Figure 13) shows, that the inclusions of each RVE either just barely do not overlap, leaving a small gap for the matrix phase, or the inclusions just barely perculate. This phenomena has a pronounced impact on the resulting effective heat conductivity. Hence, a miniscule change in the image data can result in notable variation of the conductivity tensor, which can lead to high prediction errors of the surrogate.

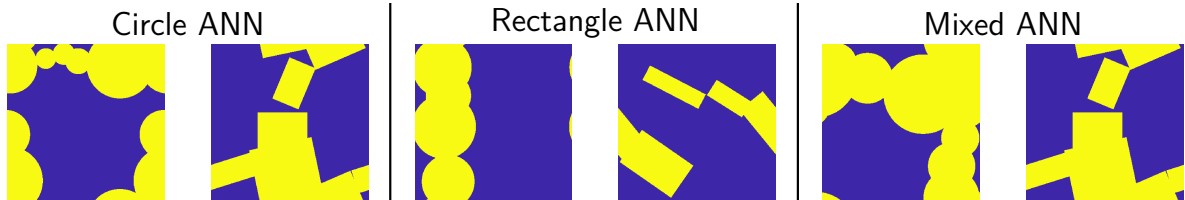

**Figure 13.** Representative volume element (RVE) with the highest prediction error for each of the ANN models given in Figure 10.

A detailed study of various ANN architectures revealed, that almost every architecture was suitable for the regression problem, e.g., an ANN with 2 hidden layers and a total of 13 hidden neurons had almost identical prediction errors as an ANN with 5 hidden layers and roughly 230 hidden neurons. The used activation functions were the sigmoid, relu, tanh and softplus, where only some combinations delivered poor results. Not a clear trend of ANN architecture and quality of prediction could be seen and, consequently, the best ANN were randomly found based on the lowest error on the test set.

The prediction accuracies for each test set of three differently trained ANNs, which have been deemed the *best*, is given in Figure 11. The training and architecture of the best ANNs in Figure 11 had the following properties:

- Circular training: $h = 23$; $11,206$ epochs; 5 hidden layers
  $\{5, 40, 77, 75, 74\}$ hidden neurons
  $\{$sigm, softplus, sigm, softplus, softplus$\}$ activation functions

- Rectangular training: $h = 29$; $1,054$ epochs; 6 hidden layers
  $\{10, 42, 56, 18, 63, 59\}$ hidden neurons
  $\{$relu, sigm, relu, softplus, tanh, tanh$\}$ activation functions

- Mixed training: $h = 26$; $6,177$ epochs; 2 hidden layer
  $\{6, 7\}$ hidden neurons
  $\{$softplus, softplus$\}$ activation function

The shown error measures (Figure 11) are evaluated for each point in the whole test set, yielding a kind of probability distribution for the prediction error. For an easier readability, the percentage mean and max errors for each of the explicitly depicted ANN are given in Table 2. Note that since the values of $\bar{\kappa}_{12}$ vary closely around 0 (Figure 4), relative errors are not sensible for the quantity of interest.

**Table 2.** Percentage errors for $\bar{\kappa}_{11}$ and $\bar{\kappa}_{22}$ given for each of the best ANNs, evaluated over the complete test set (7500 data samples).

| Trained with | Error Measures | Circles $\kappa_{11}$ | Circles $\kappa_{22}$ | Rectangles $\kappa_{11}$ | Rectangles $\kappa_{22}$ | Mixed $\kappa_{11}$ | Mixed $\kappa_{22}$ |
|---|---|---|---|---|---|---|---|
| Circles | Mean [%] | 1.58 | 1.57 | 2.60 | 2.62 | 2.11 | 2.14 |
| | Max [%] | 12.8 | 12.5 | 13.9 | 13.0 | 14.7 | 11.7 |
| Rectangles | Mean [%] | 2.68 | 2.57 | 1.60 | 1.58 | 2.14 | 2.09 |
| | Max [%] | 12.9 | 11.7 | 12.5 | 12.0 | 13.8 | 13.0 |
| Mixed | Mean [%] | 1.77 | 1.76 | 1.65 | 1.60 | 1.72 | 1.71 |
| | Max [%] | 11.7 | 14.1 | 11.6 | 10.5 | 10.4 | 12.4 |

PARAGRAPH MOVED (after the table) As a side note, a descriptor based GPM has been trained for RVEs with circular inclusions, using the average minimum distance of inclusions, average inclusion radius, number of inclusions and volume fraction as an input, achieving mean relative errors of around 5% on the circle set.

A GUI code is provided in Github, where the user can choose between the three proposed surrogate model, the input for the prediction is a $400 \times 400$ image in matrix format written in a text file or a TIFF image and the output is the prediction for the heat conduction tensor as described above. In order to compile the code, Python3 with TensorFlow is required, additional required modules are pillow, numpy and matplotlib, as well as the default modules os and tkinter. Some exemplary RVE with their respective heat conductivity are uploaded in a subfolder.

## 5. Computational Effort

For the training and the deployment of the proposed surrogate model, the computational effort can be split into online and offline part. The offline phase describes the building of the surrogate model and is obviously computationally expensive due to the iterative nature of the supervised as well as the unsupervised learning. However, since the cost of the offline phase has no impact on the actual evaluation, i.e., prediction of the surrogate model, its impact is neglectable. All of the following measured times have been documented while computing with only an AMD Ryzen Threadripper 2920X 12-Core Processor, unless stated otherwise. In order to evaluate the surrogate model in the online phase, firstly, the 2PCF of the RVE has to be computed. Therefore a FFT, complex point-wise multiplication and lastly an IFFT is performed, summing up to a computational complexity of $\mathcal{O}(2\,n\log n + n)$. Recall that $n$ is the dimension of the unreduced problem, i.e., the total number of voxels in the present study.

To derive the input for the ANN, the complexity for the computation of the reduced coefficients is $\mathcal{O}(n\,h)$ together with a computation of the volume fraction with an additional effort of $\mathcal{O}(n)$. This mounts up to a total computation effort of $\mathcal{O}(n\,(2+h) + 2\,n\log n\,)$ just to derive the input of the regression model. To give sensitivity to the computational effort, the computation of the feature vectors for one test set, i.e., 7500 images, took roughly 95 s.

As has been mentioned earlier, the ANN has been significantly faster than the GPM in the online, as well as offline phase. The training of the regression model for each number of reduced coefficients (i.e., 1–30) took roughly 12 hours for the ANN and about 31 hours for the GPM. Note that GPM has been trained in R with the code provided by [50], whereas the ANN has been implemented in Python with TensorFlow, using a Pali6GB D6 RTX 2060 GamingPro OC graphics card as well. As it is more important, in the online phase the ANN has been significantly faster than the GPM. Each prediction refers to the prediction of the three test sets, i.e., $3 \times 7500$ data points with each output being a three-dimensional vector. The prediction times for the GPM highly depends on the size of the input

vector and takes from 0.82 s (with one reduced coefficient) up to 4.1 s to predict the test sets for an input dimension of 31. In comparison, the ANN took on average roughly 0.24 s for any dimension of the feature vector.

The computational complexity of the forward propagation in the ANN is governed by the matrix multiplication of a complexity of $\mathcal{O}(n_{\text{neuron}}^2)$ and the element wise evaluation of the activation function for each neuron with the complexity $\mathcal{O}(n_{\text{neuron}})$. For a quick overview, assume that the ANN has the same number of neurons in each layer, the computational complexity amounts to $\mathcal{O}(n_{\text{layer}}(n_{\text{neuron}}^2 + n_{\text{neuron}}))$. Therefore, we have an a priori estimate of the prediction time required for the ANN.

To compute the effective heat conductivity for 7500 images using the FANS solver [3], $\approx 4000$ s were required. Note that the deployed FFT solver for the heat conductivity is intrinsically fast. The proposed method could be easily expanded to different material properties, yielding an even more significant computational speedup. Since usually $n \gg n_{\text{neuron}}$, the main computational effort lies within the computation of the feature vector, especially when considering the extension to the 3D case.

## 6. Conclusions

### 6.1. Summary and Concluding Remarks

The computational homogenization of highly heterogeneous microstructures is a challenging procedure with massive computational requirements. In the present study a method to efficiently and accurately predict the heat conductivity for any RVE with the image and no further information is proposed. Key ideas of the Materials Knowledge System (MKS) [21,32] have been adopted in the sense that a subset of the POD compressed 2-point correlation function is used to identify a low-dimensional microstructure description. In contrast to [32] the 2PCF is not truncated to a small neighborhood, but the full field information is considered. Similar to other works related to the MKS [18], a truncated PCA of the 2-point information is used to extract microstructural key features.

However, the classical truncated PCA used, e.g., in [18] is not applicable to the considered rich class of microstructures due to the high number of needed samples and the related unmanageable computational resources. Therefore, our proposal is founded on a novel incremental procedure for the generation of the RB of the 2PCF. Similar techniques have not been considered in the literature to the best of the authors' knowledge. The shifting and scaling of the images of 2PCF before entering the POD is another feature that can help in reducing the impact of the inclusion volume fraction, i.e., the shifted function has zero mean and a peak value of one. The authors would like to emphasize that such scaling is relevant in the present study where the phase volume fractions varies in a wide range.

Other than in [32] no higher-order statistics are used. This is by purpose as the selection of the relevant entries of the higher order PCF is ambiguous and a challenge in itself. Most notably it is based on a priori selections of the relevant components of the higher spatial correlations which allows for very limited insights to our understanding. Instead, the present study focuses on the variability of the input images in terms phase volume fractions in a broad range (20–80%) alongside topological variations (impenetrable, partial overlap, unrestricted placement) and different morphologies (circles and rectangles). Generally speaking, a much higher microstructural variation is accounted for, than in many previous studies. Therefore, the current study also investigates how the proposed technique and similar MKS related approach can possibly generalize towards truly arbitrary input images (e.g., stemming from 3D micrographs of real materials) and for databases containing millions of snapshots in order to build a powerful tool for material analysis and design.

In order to cope with the variability of the 2PCF, the classical truncated PCA or snapshot POD operating on a monolithic snapshot matrix during the unsupervised learning phase is replaced by novel incremental procedures for the construction of small-sized reduced microstructure parameterization. Three incremental POD methods are proposed and their results are compared regarding the

computational effort, the projection accuracy of the snapshots and the quality of the basis in view of capturing random inputs.

The learned reduced bases are used to extract low-dimensional feature vectors. These are used as inputs for fully connected feedforward Artificial Neural Networks. The ANN is used to predict the homogenized heat conductivity of the material defined by the microstructure. The mean relative error of the surrogate is well below 2% for the majority of the considered test data. This is remarkable in view of the phase contrast $R = 5$ and the particle volume fractions ranging from 0.2–0.8, as well as morphological and topological variations. Further, an immense speedup in computing time is achieved by the surrogate over FE or FFT simulations (factors around 40 without tweaking the projection operation).

Importantly, the presented methodology can immediately be adopted to different physical settings such as thermo-elastic properties, fluid permeability, dielectricity constants, etc. The same holds for three-dimensional problems. However, the limited number of samples in 3D could be problematic as more features are likely required to attain a sufficiently accurate RB.

### 6.2. Discussion and Outlook

A weakness of the current approach remains the computational complexity of the method: Although the feature vector is rather low-dimensional, it requires the evaluation of the 2PCF using the FFT which is of complexity $\mathcal{O}(n \log(n))$ where $n$ is the number of pixels/voxels in the image. In order to extract the reduced coefficient vector from the 2PCF, the latter must be projected onto the RB. This operation scales with $\mathcal{O}(n\,h)$. These two operations are at least linear to the number of pixels or voxels of the image which can be critical, especially in three-dimensional settings. Consequently, the computational effort of the feature vector computation heavily out-weights the computational complexity of the regression model as can readily be seen from the provided timings (95 s vs. 0.08 s for the ANN for 7500 predictions). In the future, optimizations, e.g., in the spirit of reduced cubature rules [51], will be explored to render the overall computation more efficient in view of 3D microstructures at resolutions of $512^3$ and beyond.

Another extension of the current scheme could account for variable phase contrast $R$ which was fixed as $R = 5$ in this work. In particular, higher phase contrasts should be explored. Preliminary investigations state that the accuracy of the machine learned surrogate deteriorates considerably for a high phase contrast of $R = 1/100$. The source of error and the possible measures to cope with extreme contrasts ($R \ll 1$ and $R \gg 1$) in the data-driven model should be studied in the future. Thereby, the dimension of the feature vector must increase, even beyond the 2PCF. This could possibly lead to a data scarcity dilemma: The number of input samples for the supervised learning should grow exponentially with the dimension of the feature vector. However, this is not realizable in practice due to limited computational resources. With the goal of predictions for nearly arbitrary 3D microstructures in mind, in the authors' opinion this dependence is the most pronounced short-coming of the method and future studies should focus on limiting the number of required input samples in order to fight the curse of dimensionality as more reduced coefficients require an exponential growth in the available data, making the offline procedure unaffordable, today.

Advantages of the current scheme comprise the independence of the underlying simulation scheme. This does allow for heterogeneous simulation environments, the use of commercial software, multi-fidelity input data and blended sources of information (e.g., in silico data supported by experimental results).

**Author Contributions:** Conceptualization, J.L. and F.F.; Data curation, J.L.; Formal analysis, F.F.; Funding acquisition, F.F.; Investigation, J.L. and F.F.; Methodology, J.L. and F.F.; Project administration, F.F.; Resources, F.F.; Software, J.L.; Supervision, F.F.; Validation, J.L.; Visualization, J.L. and F.F.; Writing—original draft, J.L. and F. F.; Writing—review & editing, J.L. and F.F.

**Funding:** This research was funded by Deutsche Forschungsgemeinschaft (DFG) within the Emmy-Noether programm under grant DFG-FR2702/6 (contributions of F.F.).

**Acknowledgments:** Support from Mauricio Fernández on the implementation and layout for the training of the Artificial Neural Networks using Google's TensorFlow is highly appreciated. Stimulating discussions within the Cluster of Excellence SimTech (DFG EXC2075) on machine learning and reduced basis methods are highly acknowledged. Further, the authors would like to thank the three anonymous reviewers for their remarks (particularly in view of the GPM method) which helped in improving the quality of the manuscript.

**Conflicts of Interest:** The authors declare no conflict of interest.

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
