# Peer review of "Data-Driven Microstructure Property Relations"

_mca, doi:10.3390/mca24020057_

Round 1

Author Response

Thank you for the thorough review. We have summarized all responses into the attached document.

Reviewer 2 Report

This paper develops three incremental snapshot proper orthogonal

decomposition methods focused on mapping 2-point correlation function

representations for discrete microstructures to low-dimensional feature

vectors.  A multilayer perceptron model is used to demonstrate how these

features can be used to map microstructure inputs to mean-field

properties via the resulting POD features (using effective heat

conductivity as a case study).

I think it's a nice paper that could benefit from a bit of additional

discussion surrounding the supervised learning section.

1 Supervised learning section

═════════════════════════════

  The neural network case study is very nice, but it seems that any

  other supervised regression model could take the place of the neural

  network in this work.

  I think it would be nice to know how much the neural network

  contributes to the predictive performance of the model, and how much

  can be attributed to the POD.  One means of assessing this could be to

  compare the neural network results with a linear model as a baseline

  to establish how difficult the regression problem is.

  Related: the 'modern' (or perhaps trendy) deep learning approach to

  this problem setting would be to feed the raw microstructure data into

  a large convolutional neural network, abandoning spatial statistics

  entirely.

  It would be helpful if you explicitly discuss the tradeoffs in terms

  of sample complexity, interpretability, etc between this approach and

  your approach, and highlight the benefits of the 2-point statistics +

  POD + regression model approach in the context of application to

  practical materials problems.

2 training hyperparameters?

═══════════════════════════

  Please discuss the range of hyperparameters used in tuning the neural

  network model.  Documentation of these values also will improve

  accessibility and reproducibility of the work.

  • initial learning rate and schedule (if any) used with the Adam

    optimizer

  • early stopping patience (how many iterations/epochs with no

    improvement do you allow before stopping optimization?)

  • what ranges of network depth and width do you consider?

  • do you use any regularization? (batch normalization, dropout, weight

    decay, etc?)

  How many trials are performed in your random hyperparameter search?

3 noise in Fig 9 is concerning

══════════════════════════════

  The amount of noise in Figure 9 as the input dimensionality increases

  is somewhat concerning.  For each point in a single curve, is the

  neural network architecture constant, or is an independent

  hyperparameter search performed for each input feature size?

  If the latter, then the reason for the high variance could be due to

  poorly converged hyperparameter search.  If the former, then maybe

  this is just a result of overfitting.

4 use the training set to make a learning curve?

════════════════════════════════════════════════

  on page 18 you suggest that the apparent overfitting behavior that you

  observe could be a result of small training set size.  You could

  quantitatively assess this by constructing a learning curve (i.e. plot

  the validation error vs (log) training set size to see whether the

  validation error has converged).

5 Thank you for adhering to community best practices and openly sharing your code

═════════════════════════════════════════════════════════════════════════════════

Author Response

(The authors gave the same response as above.)

Reviewer 3 Report

Summary

In this paper, the authors build a surrogate model (neural network, NN) that links reduced-order representation of two-phase microstructures to their homogenized property (in particular, effective heat conductivity). Microstructure morphology is characterized via 2-point correlation function which is then represented in the reduced-dimension via SVD.

Comments

1.     There are many similar works on this topic and hence the authors have to make their “unique” contribution clear. There are multiple statements on this issue which are not quite accurate. For example, “The goal in the present study is to make accurate image based predictions for RVEs spanning large subsets, e.g., in terms of volume fraction, morphological and topological variations, of microstructure materials” has been done before in:

a.      Bessa, M. A. et al. A framework for data-driven analysis of materials under uncertainty: Countering the curse of dimensionality. Comput Method Appl M 320, 633-667, doi:10.1016/jNaNa.2017.03.037 (2017).

b.     Xu, H. Y., Li, Y., Brinson, C. & Chen, W. A Descriptor-Based Design Methodology for Developing Heterogeneous Microstructural Materials System. J Mech Design 136, 051007, doi:Artn 051007

c.      10.1115/1.4026649 (2014).

2.     There are many other methods to generate synthetic microstructures. What is the unique feature of this approach? For a list of methods, see this review paper:  Bostanabad, R. et al. Computational microstructure characterization and reconstruction: Review of the state-of-the-art techniques. Prog Mater Sci 95, 1-41, doi:10.1016/j.pmatsci.2018.01.005 (2018). Example methods also include:

a.      Groeber, M. A. & Jackson, M. A. DREAM. 3D: a digital representation environment for the analysis of microstructure in 3D. Integrating Materials and Manufacturing Innovation 3, 1 (2014).

b.     Bostanabad, R., Bui, A. T., Xie, W., Apley, D. W. & Chen, W. Stochastic microstructure characterization and reconstruction via supervised learning. Acta Materialia 103, 89-102 (2016).

c.      Kumar, A., Nguyen, L., DeGraef, M. & Sundararaghavan, V. A Markov random field approach for microstructure synthesis. Model Simul Mater Sc 24, 035015, doi:Artn 03501510.1088/0965-0393/24/3/035015 (2016).

3.     There are some errors in the manuscript that need to be fixed. For example: “… the principal scores are used to in a polynomial regression …“ does not read well.

4.     The arrows in Fig. 2 are not clear.

5.     What is the application of this work? For example, what can the fitted NN be used for? Obviously this cannot be directly used for design as the reduced-representation does not have a physical meaning.

6.     Why have the authors used NN? The dataset size (15 K) is relatively small can be easily handled via simpler (and more accurate and more user-friendly) methods such as Gaussian processes. See: Bostanabad, R., Kearney, T., Tao, S., Apley, D. W. & Chen, W. Leveraging the nugget parameter for efficient Gaussian process modeling. Int J Numer Meth Eng 114, 501-516, doi:10.1002/nme.5751 (2018).

Author Response

(The authors gave the same response as above.)

Round 2

Reviewer 1 Report

The authors accounted for previous remarks and have updated the manuscript in accordance with my review recommendations. Hence, I submit the paper for publication "as is".

Reviewer 3 Report

The authors have addressed the comments. Two notes:

Did the authors "use" GPM are "developed" their own version?

There are some minor grammar issues but hopefully the authors will catch them before submission.